# Management of Early-Stage Vulvar Cancer

**DOI:** 10.3390/cancers14174184

**Published:** 2022-08-29

**Authors:** Priscila Grecca Pedrão, Yasmin Medeiros Guimarães, Luani Rezende Godoy, Júlio César Possati-Resende, Adriane Cristina Bovo, Carlos Eduardo Mattos Cunha Andrade, Adhemar Longatto-Filho, Ricardo dos Reis

**Affiliations:** 1Molecular Oncology Research Center, Barretos Cancer Hospital, São Paulo 14784-400, Brazil; 2Department of Prevention Oncology, Barretos Cancer Hospital, São Paulo 14784-400, Brazil; 3Department of Prevention Oncology, Barretos Cancer Hospital, Mato Grosso do Sul 79085-040, Brazil; 4Department of Gynecologic Oncology, Barretos Cancer Hospital, São Paulo 14784-400, Brazil; 5Barretos School of Health Sciences, Dr. Paulo Prata-FACISB, Barretos, São Paulo 14785-002, Brazil; 6Medical Laboratory of Medical Investigation (LIM) 14, Department of Pathology, Medical School, University of São Paulo, São Paulo 01246-903, Brazil; 7Life and Health Sciences Research Institute (ICVS), School of Medicine, University of Minho, 4710-057 Braga, Portugal; 8ICVS/3B’s—PT Government Associate Laboratory, 4710-057 Braga, Portugal; 9ICVS/3B’s—PT Government Associate Laboratory, 4805-017 Guimarães, Portugal

**Keywords:** vulvar cancer, vulvar neoplasms, diagnosis and staging, sentinel lymph node

## Abstract

**Simple Summary:**

Vulvar cancer is a rare gynecological malignancy that affects mainly postmenopausal women. Recently, however, an alarming increase in the rates among young women has been observed due to human papillomavirus infection. The standard treatment for vulvar cancer is surgery with or without radiotherapy as adjuvant treatment. In recent decades, sentinel lymph node biopsy has been included as part of the surgical treatment. Thus, our objective was to review and discuss the advances found in the literature about early-stage vulvar cancer. For this, we searched PubMed for publications in the English language. Relevant articles, such as the GROINS-V studies, and the GOG protocols, are presented in this review exhibiting the evolution of early-stage vulvar cancer treatment and the decrease in surgical morbidity rates.

**Abstract:**

Vulvar cancer is a rare gynecological malignancy since it represents 4% of all cancers of the female genital tract. The most common histological type is squamous cell carcinoma (90%). This type can be classified into two clinicopathological subtypes according to the etiology. The first subtype is associated with persistent human papillomavirus infection and is usually diagnosed in younger women. The second subtype is associated with lichen sclerosus condition, and in most cases is diagnosed in postmenopausal women. Currently, an increase in first subtype cases has been observed, which raised the concern about associated mortality and treatment morbidity among young women. Vulvar cancer treatment depends on histopathology grade and staging, but surgery with or without radiotherapy as adjuvant treatment is considered the gold standard. In recent decades, sentinel lymph node biopsy has been incorporated as part of the treatment. Therefore, we sought to review and discuss the advances documented in the literature about vulvar cancer focusing on the treatment of early-stage disease. Relevant articles, such as the GROINS-V studies and the GOG protocols, are presented in this review. Additionally, we discuss key points such as the evolution of treatment from invasive surgery with high morbidity, to more conservative approaches without compromising oncologic safety; the role of sentinel lymph node mapping in the initial staging, since it reduces the complications caused by inguinofemoral lymphadenectomy; the recurrences rates, since local recurrence is common and curable, however, groin-associated, or distant recurrences have a poor prognosis; and, finally, the long-term follow-up that is essential for all patients.

## 1. Introduction

According to the Surveillance, Epidemiology, and End Results (SEER) Database, vulvar cancer accounts for 4% of all female genital tract cancers, and is considered as a rare gynecologic malignancy [1]. The International Agency for Research on Cancer (IARC), estimates that approximately 45,000 new cases of vulvar cancer are diagnosed each year, of which 50.1% are in high-income countries [2]. This neoplasia is responsible for approximately 17,000 deaths per year, with the majority in high-income countries (40.8%) [3].

Ninety percent of vulvar cancers are squamous cell carcinoma (SCC) [4]. Rarer histologic subtypes include basal cell carcinoma, verrucous carcinoma, Bartholin’s gland adenocarcinoma, extramammary Paget’s disease, and vulvar melanoma [5].

Vulvar SCC has classically been considered a disease of older and postmenopausal women, associated with lichen sclerosus and other vulvar inflammatory epithelial conditions that may be related to differentiated vulvar intraepithelial neoplasia [6]. Although the median age at diagnosis is 69 years [1], recent evidence shows that, worldwide, an increase in vulvar SCC cases among young women has been observed [7]. This intensification is related to persistent high-risk human papillomavirus (HPV) infection that has led to an increase in HPV-associated high-grade vulvar squamous intraepithelial lesions. In this context, approximately 45% of all vulvar SCC cases are thought to be caused by HPV infection [8,9]. In addition to HPV infection, other risk factors are involved in these increased rates of vulvar SCC, such as ethnicity, smoking, inflammatory conditions of the vulva, and prevalence of the immunodeficiency virus (HIV) [7,10].

In an epidemiological study, Kang et al. observed an increase of 14% in the global incidence of vulvar cancer in the 13 high-income countries. In addition, they found that this augment was not distributed consistently across age groups, showing a significant increase of 38% in the overall incidence in women under 60 years of age and no significant increase in women over 60 years of age [7].

Although vulvar cancer is a rare gynecological neoplasm in older women, SCC is the most common histological type. In recent years, an increase in this neoplasm in young women has been observed due to persistent HPV infection, leading to a concern about mortality and morbidities in treating these patients. Considering the diagnosis and staging of the disease, surgical treatment, lymph node evaluation, adjuvant treatment, and recurrence, the aim of this paper was to provide recommendations and advances in the treatment of early-stage vulvar cancer.

## 2. Vulvar Anatomy

Anatomically, the female external genitalia comprises the vulva and the pubic region or pubic mound. The vulva is in the anterior triangle of the perineum and is composed of the labia majora and minora, clitoris, vaginal vestibule bulb, and the minor (Skene’s glands) and major (Bartholin’s glands) vestibular glands (Figure 1). The internal and external pudendal arteries are responsible for most of the vulvar blood supply. The ilioinguinal, genitofemoral, and pudendal nerves are responsible for innervating the vulvar tissue [11,12].

The vulvar lymphatic drainage is conducted by the inguinal lymph nodes. The superficial inguinal nodes cross the cribriform fascia and correspond to the first line in the lymph chain. The deep inguinal nodes are the second in the lymph chain, followed by the external iliac nodes and ending in the paraaortic nodes. Drainage is distributed within the inner third of the vaginal tube and the outer part of the anus (below the anal sphincter). Unilateral or bilateral lymphatic drainage can occur depending on the size, location, and proximity of the primary tumor to the midline; but, if the lesion is close to or on the clitoris, lymphatic drainage might be made by the iliac region [9,13] (Figure 2).

## 3. Prevention

### 3.1. Vaccination (Primary Prevention)

Persistent HPV infection, particularly type 16, has been associated with high-grade squamous intraepithelial lesions (HSIL) and vulvar SCC [14,15]. Since the introduction of the HPV vaccine as a primary preventive strategy for cervical cancer, several pieces of evidence have been carried out to verify the effectiveness of the vaccine against non-cervical female genital neoplasms associated with HPV infection [15,16,17,18,19,20,21]. These studies demonstrated that vaccination against HPV is also an effective preventive measure against vulvar cancer, presenting a promising future reduction in the incidence. It is estimated that the vaccines will be able to prevent 70% of vulvar cancer cases related to HPV infection [15,16,17,18,19,20,21].

### 3.2. Screening (Secondary Prevention)

No evidence of specific screening tests for vulvar cancer has been reported to the best of our knowledge. Women with lichen sclerosus—an inflammatory disease of unknown etiology, probably autoimmune—should perform self-examination in search of suspicious lesions [22,23]. In addition, for women who have intraepithelial lesions of the cervix, vagina, or anus, it is mandatory to inspect the vulva as part of their colposcopic follow-up appointments [24].

## 4. Precursor Lesions

In the early 1980s, the term vulvar intraepithelial neoplasia (VIN) was created. In 2012, the Lower Anogenital Squamous Terminology (LAST) Project merged the terminology for all squamous lesions associated with HPV infection and recommended the use of the terms low-grade squamous lesion (LSIL) and high-grade squamous lesion (HSIL) on the vulva. In 2014, the World Health Organization (WHO) subdivided squamous intraepithelial lesions of the vulva into LSIL, HSIL, and differentiated VIN (Table 1) [25].

In 2015, the International Society for the Study of Vulvovaginal Disease (ISSVD) accepted and approved the new terminological classification. This term is currently accepted for vulvar lesions with histological features of squamous epithelial atypia and squamous cell carcinoma in situ (CIS) of the vulva [26].

**Table 1 cancers-14-04184-t001:** Vulvar intraepithelial neoplasia (VIN) terminology changes [27,28,29,30].

ISSVD 1986	ISSVD 2004	LAST 2012/WHO2014
VIN I	Flat condylomata or HPV effect	LSIL
VIN II and III	VIN, usual type: (i) VIN, warty type (ii) VIN, basaloid type (iii) VIN, mixed	HSIL
Differentiated VIN	VIN, differentiated type	Differentiated VIN

ISSVD: International Society for the Study of Vulvar Disease; LAST: Lower Anogenital Squamous Terminology Project; VIN: vulvar intraepithelial neoplasia; LSIL: low-grade squamous intraepithelial lesions; HSIL: high-grade squamous intraepithelial lesions; dVIN: differentiated VIN.

The precursor lesion of the SCC is the VIN. According to Abell, VIN is classified into two clinicopathological subtypes: usual VIN (uVIN) and differentiated VIN (dVIN) [31]. Usual VIN is associated with HPV infection, has HSIL as a precursor lesion, and is less likely to progress to SCC [25], while dVIN is not related to HPV infection, but arises from chronic dermatoses, mainly lichen sclerosus and lichen planus [32] (Figure 3).

Histologically, uVIN progresses to basaloid or verrucous SCC, and dVIN progresses to keratinized SCC [33]. Regarding immunohistochemistry (IHC), uVIN is usually positive for p16INK4a and negative for p53, in contrast, dVIN is negative for p16INK4a and positive for p53 [34]. In the translational sub-study, AGO-CaRE-1, Woelber et al. evaluated the role of HPV and p53 status in patients with primary vulvar SCC. Four hundred and eleven samples were analyzed by IHC and the authors observed three expression subgroups relevant to vulvar SCC: p16−/p53+ (*n* = 163), p16+/p53− (*n* = 132), and p16−/p53− (*n* = 116). When performing multivariate analysis, the p16+/p53− subgroup showed a better prognosis compared with the other subgroups (hazard ratio (HR) = 0.66). With this, the authors conclude that there is a third subgroup of expression relevant to vulvar SCC, p16−/p53−, which has an intermediate prognosis and still needs to be better characterized [35].

## 5. Diagnosis

As vulvar resemble inflammatory epithelial conditions, it is common that most cases of vulvar cancer are initially diagnosed as inflammatory conditions. The initial misdiagnosis leads to a delay in treatment (on average 2 years), worsening their prognosis. For this reason, dermatologists must be familiar with vulvar malignancies, because early diagnosis leads to a more favorable prognosis [36,37,38].

Most cases of vulvar cancers are associated with the skin of the lips, whereas neoplasms that arise in the clitoris and vestibular glands are extremely rare [39]. Vulvar cancer may present as a palpable lump, a visible lesion, and/or a raised, flat ulcerated, or warty mass on the vulva. The lesions may be skin-colored, white, erythematous, or pigmented. Some patients are asymptomatic, but others present with pruritus, irritation, pain, dysuria, burning, or bleeding, especially in cases of invasive disease [40,41].

Any suspicious lesion that is found on the vulva should be biopsied. Benign lesions are considered common dermatopathology, such as atopic dermatitis, contact dermatitis, psoriasis, pemphigus vegetans, mycosis fungoides, lichen chronicus simplex, lichen sclerosus, or lichen planus. Other benign vulvar lesions can appear through infections such as candidiasis, herpes, or vestibulitis. The biopsy is essential because VIN is a pre-invasive lesion, but it can resemble these more common skin conditions mentioned above or even other atypical malignancy lesions such as extramammary Paget’s disease, and basal cell carcinoma, or melanoma [42].

Extramammary Paget’s disease may have a nonspecific identification, leading to an average delay of two years for its diagnosis, usually after the failure of steroids or topical antifungals. In the same way, basal cell carcinoma can also be nonspecifically identified as a painless visible tumor driving to incorrect diagnosis and treatment [43]. For vulvar melanomas, the diagnosis can be hindered since about 25% of them are amelanotic [44].

Normally, vulvar cancer is detected by general gynecologists or dermatologists, but due to its rarity, women should be referred to cancer centers for treatment since it must be multidisciplinary, involving oncological gynecologists, dermatologists, and pathologists [45].

Clinical and morphological examination is part of the diagnostic investigation, but histopathology is a unique gold standard peremptorily accepted for the diagnosis of any suspicious lesion. On clinical examination, it is possible to observe an erythematous lesion, plaques or ulcers, a scaly plaque, or an ill-defined mass. Usually, lesions that look similar to cauliflower are verrucous carcinoma. As anticipated, any suspicious lesion must be carefully investigated and biopsied. In addition, pelvic examination and colposcopy of the vulva and vagina should be performed [46]. In cases of vulvar melanoma, the ABCDE rule—a dermatological acronym for asymmetry, border irregularity, color, diameter, and evolution—can help in the diagnosis [47].

Vulvar lesions must be confirmed histologically and are classified as vulvar cancer only when the primary site of the tumor is the vulva. Thus, any tumor involving the vulva and vagina, crossing the hymenal ring, should be considered vulvar cancer, except for secondary tumors of genital and extragenital sites. Vulvar melanoma may be reported separately and requires histological confirmation [48].

The nodal involvement can be unilateral or bilateral, depending on the location—closer to the clitoris or midline—and tumor size. The first site of spread is the inguinal and femoral lymph nodes, followed by the pelvic lymph nodes. Patients who have metastases to the pelvic lymph nodes—external, hypogastric, obturator, and common iliac—or extrapelvic spread are classified as stage IV vulvar cancer [49].

Meticulous mapping of all biopsy sites is critical, and should contain the precise anatomical position as well as the location on a clock concerning the distance from the midline and vaginal introitus; this location is crucial for planned surgical management. To assess the extent of the disease, computed tomography (CT) or positron emission tomography (PET/CT) imaging, and magnetic resonance imaging (MRI) are performed and if there is evidence of bladder or rectal involvement, cystoscopy and proctoscopy should be performed [46].

In the case of extramammary Paget’s disease, screening for other neoplasms, such as genitourinary, gastrointestinal, and breast cancer should be performed, considering that the most common presentation in secondary vulvar Paget’s disease is anorectal and urothelial adenocarcinomas [50].

## 6. Staging

Vulvar cancer staging is determined according to the International Federation of Gynecology and Obstetrics (FIGO) (Table 2) and is applicable in most vulvar neoplasms, except vulvar melanoma. Since 1988, vulvar cancer has been surgically staged and the final diagnosis is based on a complete histopathological evaluation of the surgical specimen (vulva and lymph nodes) [49].

Before planning the surgical treatment, it is necessary to perform a pelvic exam with inspection of the vagina, cervical cytology, and colposcopy of the vulva, vagina, and cervix, to discard other HPV-related pre-invasive lesions or cancers. HPV testing may also be requested. Additionally, full blood, biochemical evaluation, and HIV testing are indicated. If concerned, radiological exams, chest X-ray, MRI, and PET/CT of the pelvis and groins may be helpful for lymph node evaluation and further surgical treatment planning [39,46,48,51,52].

## 7. Treatment

The treatment of vulvar cancer depends mainly on histology and staging, but other factors influence the management, such as the patient’s age, performance status, and the presence of comorbidities. Surgical excision is the gold standard therapy for vulvar cancer, although radiotherapy and chemotherapy are effective alternatives, particularly in advanced tumors and in those tumors in which exenteration would be essential to obtain adequate surgical margins [53]. Other treatments are performed in metastatic, palliative, or vulvar melanoma cases, such as chemotherapy and immunotherapy [46,54]. All treatment management must be individualized and carried out in specialized cancer centers with multidisciplinary teams. Additionally, offering psychosexual attendance for all women during the diagnosis, treatment, and post-treatment of pre-invasive or invasive vulvar disease is highly recommended [54].

### 7.1. Surgical Management

Surgical treatment must be performed individually to maintain the surgery as conservative as possible and ensure oncological safety [48,55,56]. When the surgical option is considered, proper management of the primary lesion and groin lymph nodes must be analyzed separately to maximize the chances of cure and minimize treatment-related morbidity [46,48,54,55,56,57].

#### 7.1.1. Microinvasive (Stage IA)

Microinvasive vulvar carcinoma (stage IA) is defined as a lesion with a diameter of 2 cm or less and an invasion depth of 1 mm or less. Depth of invasion is measured across the epithelium–stromal junction of the most adjacent, dysplastic, tumor-free, superficial dermal papilla to the deepest point of invasion [58,59]. This type of lesion should be treated with wide, radical local excision, and inguinal lymph node evaluation is not necessary [55].

#### 7.1.2. Early-Stage

Early-stage vulvar cancers are those tumors confined to the vulva without suspicious lymph nodes, either on clinical examination or cross-sectional radiological evaluation [48,60].

Formerly, in the mid-20th century, early-stage patients were treated with radical vulvectomy, including, “en bloc” bilateral pelvic and inguinofemoral lymphadenectomy, but the survival rate was 75% at 5 years [61,62]. For this reason, and the severe post-surgical complications, such as infection, wound breakdown, lymphedema, and psychosexual disorders [61,63,64], the performance of radical vulvectomy on lesions up to 2 cm in diameter was contested about 40 years ago [65,66]. The concern for more conservative procedures that would result in more vulvar recurrence is explained by the concept of a cancerization field [67].

Currently, the gold standard treatment of early-stage vulvar cancer is wide and radical local excision of the tumor [68]. This procedure is more conservative than, and equally as effective as radical vulvectomy, preventing local recurrence and substantially decreasing the psychosexual morbidity of the treatment [48,69].

#### Role of Surgical Margins

Historically, surgical margins of 1 cm or even 2 cm were recommended to achieve pathologic tumor-free margins of at least 8 mm (allowing for shrinkage of the fixed tissue) [70]. This recommendation has been challenged and it is recognized that the vast majority of recurrent vulvar cancers are new tumors that have arisen in the surrounding abnormal tissue and not local recurrence due to inadequate margins [71].

Te Grootenhuis et al. evaluated the incidence of local recurrence in vulvar cancer regarding pathological margins free of tumor and/or precursor lesion. They re-examined slides of 287 patients surgically treated at two Dutch specialized centers between 2000 and 2010, with a mean follow-up of 80 months. In addition to analyzing classical pathologic parameters—such as margin status and tumor factors—the authors also investigated the presence of premalignant lesions in the margin of the pathologic specimen. In the multivariate analysis, the distance of the pathologic tumor-free margin did not influence the risk of local recurrence (42.5%) after 10 years of treatment at three different cut-off points (<8 mm, 5 mm, or 3 mm). However, patients with lichen sclerosus and dVIN lesions at the resection margin or adjacent to the tumor had a higher local recurrence rate (76.4%) after 10 years of treatment. Patients with premalignant lichen sclerosus and dVIN lesions in the pathologic tissue showed an increased local recurrence rate indicating a possible cancerization effect that influenced this increased risk of local recurrence [72].

In another retrospective study, Bedell et al. evaluated whether re-excision or adjuvant radiation in patients with early-stage vulvar cancer with positive or close (<8 mm) surgical margins improved local recurrence-free survival. Medical records of 150 patients examined between 1995 and 2017 with a mean follow-up of 25 months were revised and analyzed. Approximately 31% (*n* = 47) of patients had positive or close (<8 mm) margins. Approximately, 45% of patients underwent re-excision (*n* = 17) or adjuvant radiotherapy (*n* = 26). Local recurrence-free survival and overall survival were similar between patients who underwent re-excision or adjuvant radiation (*p* = 0.10) and patients who did not receive additional therapy (*p* = 0.16). The authors concluded that re-excision and adjuvant radiation after primary surgical resection in patients with early-stage vulvar cancer with positive or close surgical margins (<8 mm) did not improve local recurrence-free survival or overall survival, but there was a trend toward improved local recurrence-free survival in patients who underwent re-excision or adjuvant radiation [73].

The studies by Te Grootenhuis et al. and Bedell et al. present important considerations for clinical recommendations in the treatment and surveillance of early-stage vulvar cancer: (i) the use of a smaller tumor-free margin (≥3 mm) reduces patient exposure to potentially harmful and often mutilating therapies; (ii) excision of lesions suspicious for dVIN during primary tumor resection and re-excision if dVIN is present in the pathological margin is recommended; (iii) although the morbidity associated with radical surgery was significant, it is prudent to constantly look for practice recommendations that may not reduce the risk of local recurrence after surgical treatment of vulvar carcinoma, especially in patients with dVIN in the margin [72,73]. We must understand that both studies were retrospective cohorts, with limitations that this kind of study usually presents.

An Australian study retrospectively evaluated the survival rates after conservative vulvar resection and the relationship between vulvar recurrences and surgical margins. In this study, 345 women treated with surgery for SCC were included, and the average follow-up of these patients was 93 months. The 5-year disease-specific survival rate was 86%. There were 78 vulvar recurrences, of which approximately 42% were at the primary site and approximately 58% were at a remote site. Of these vulvar recurrences, 27 (34.6%) were in patients with margins < 8 mm and 51 (65.4%) were in patients with margins ≥ 8 mm. On multivariate analysis, patients with margins < 5 mm had a significant increase in all vulvar recurrences (HR = 2.29) and recurrences at the primary site (subdistribution hazard ratio (SHR) = 15.19). Patients with a margin of 5 mm to <8 mm had a higher risk of recurrence at the primary site (SHR = 8.92) and a lower risk of recurrence at a remote site (SHR = 0.08). Patients with excision margins <8 mm treated with radiotherapy or re-excision had a lower risk for all vulvar recurrences (HR = 0.15). The authors concluded that: (i) an excision margin of at least 1 cm is recommended, which is equivalent to a histopathological margin of 8 m; and, (ii) patients with a histopathological margin < 5 mm should be treated with radiotherapy or surgical re-excision [74].

Although Barlow et al. showed that a more conservative approach to vulvar resection is an advantage [74], the exact extent of the safe tumor-free surgical margin has been disputed. Some studies showed that margin distance is not a predictor of vulvar recurrences [72,73,75,76,77,78,79,80]. Conversely, Yang et al. showed the opposite, that local recurrences were associated with tumor-free margins <8 mm [81].

In the literature, there is a wide range of divergent data about the meaning of surgical margins and, consequently, about surgical recommendations, as well. Preti et al. and Nooij et al. suggested that positive tumor margins are the only risk factor for recurrence [82,83]. Woelber et al., on the other hand, proposed that regardless of tumor-free margin it is critical to achieve a complete tumor resection [78]. Groenen et al. suggested removing just enough surrounding tissue [75], in contrast, other studies recommended that tumor-free margins should be 2 mm [79], 5 mm [84,85], or ≥8 mm [72,74,83,86,87,88]. Table 3 presents the findings of the main studies on surgical margins.

The European Society of Gynecologic Oncology (ESGO) guideline [54] was based on data from studies that discussed the influence of resection distance on tumor-free margin [72,74,87,88,89]. According to the current ESGO [54] and National Comprehensive Cancer Network (NCCN) [90] guidelines, a surgical excision margin of at least 1 cm of macroscopic skin lesion (which translates to a histopathological margin of 8 mm) is advocated, but a narrower margin may be acceptable when the tumor is close to the midline—clitoris, urethra, and anus—preventing its functionality [54,90].

Tumor-free margins reduce the risk of recurrence [72]. Therefore, radical resection should be performed to obtain a tumor-free margin and not compromise urinary continence. The margin excision should extend to the deep perineal fascia of the urogenital diaphragm and, if necessary, 1 cm to the distal urethra should be excised [48,55]. In most vulvar cancers, primary closure is feasible, but reconstructive surgery is recommended in cases of closure of large defects and maintenance of vaginal function. When reconstructive surgery is necessary, the most commonly used flaps are V–Y flap, rhomboid flap, and gluteus maximus myocutaneous flap [91].

Regarding verrucous carcinoma, local excision is usually effective; however, in advanced disease, it may require radical resection [92]. In extramammary Paget’s disease, the standard treatment is local excision, but as a multifocal disease is common, recurrences are often identified. Therefore, if the invaded area is greater than 1 mm, inguinal lymphadenectomy should be considered [93]. For vulvar melanoma, such as cutaneous melanoma, wide local excision with tumor-free margins is also postulated, as radical surgery does not improve survival and is related to increased morbidity [94].

### 7.2. Management of Inguinal Lymph Nodes

The most important factor in reducing mortality from early-stage vulvar cancers is the proper management of inguinal lymph nodes since, despite the use of multimodal therapies, recurrences and metastasis are related to very high mortality rates [95,96]. Isolated inguinal lymph node dissection is associated with a higher incidence of groin recurrence, therefore, both inguinal and femoral lymph nodes must be removed. Consequently, current treatment involves resection of the primary tumor and lymph nodes separately [46].

For patients with tumors located <2 cm from the midline, the evaluation of the inguinofemoral lymph nodes has to be bilateral, either by systematic inguinofemoral lymphadenectomy or by sentinel lymph node biopsy. However, in patients with tumors located ≥2 cm from the midline, the evaluation of inguinofemoral lymph nodes can be ipsilateral [90] (Figure 4). According to the GROINSS-V II study, when lymph node metastasis occurs on only one side, the treatment management is related to the size of the metastasis. If micrometastases (≤2 mm) are present, the patient should be treated with adjuvant radiotherapy—total dose of 50 Gy—but if macrometastases (>2 cm) are present, the treatment continues to be systematic inguinofemoral lymphadenectomy (Figure 5). A third study of GROINSS-V III is ongoing (NCT05076942) and will evaluate whether in cases of macrometastases (>2 mm) chemotherapy concomitant to inguinofemoral radiotherapy—total dose of 56 Gy—should be indicated for treatment efficacy [97].

All patients with IB stage or resectable stage II vulvar cancer should undergo an inguinofemoral lymphadenectomy [49]. The risk of recurrence remains uncertain regarding the optimal number of resected lymph nodes [98]. The standard of care for radical vulvectomy with bilateral inguinofemoral lymphadenectomy remains the three-incision technique as it has good locoregional control and acceptable surgical morbidity [99].

### 7.3. Role of Sentinel Lymph Node

Sentinel lymph node (SLN) biopsy in vulvar cancer was first described by Levenback et al. [100]. It is a less invasive technique than complete lymphadenectomy and significantly reduces the risk of lymphedema, wound infection, and dehiscence, and does not compromise survival rates or groin recurrence rates [88,101,102]. The SLN biopsy represents the selective mapping of the first lymph node drained from a malignant tumor. At least theoretically, the first lymph node can harbor metastatic cancer cells in the lymph node chain. Additionally, the SLN biopsy can detect low-volumes of metastases (isolate tumor cells ≤0.2 mm; micrometastases >0.2 mm to ≤2 mm) in patients where hematoxylin–eosin is negative for macrometastases (>2 mm) [101,103], and is safe in patients with unifocal vulvar SCC with tumors measuring up to 4 cm and clinically negative lymph nodes [101,104], with a low false-negative rate (3.7%) [103]. Therefore, detection, evaluation, and selective removal of SLN may be a beneficial oncologic procedure to decrease surgical morbidity related to extensive nodal dissection in patients with negative lymph nodes [105]. The SLN concept proved to be safe and feasible in other neoplasms, such as breast [106,107,108], melanoma [109,110,111], and some gynecological cancers, including vulvar cancer [101,103,112]. Studies performed through SLN biopsy in patients with early-stage vulvar cancer demonstrated that SLN mapping is safe, presents high rates of cancer detection, and is highly effective when performed by experienced surgeons [13,88,101,113,114].

The conventional technique for SLN mapping in vulvar cancer consists of a peritumoral injection of technetium-99m nanocolloid (TC-99) before surgery, combined with an intraoperative intradermal injection of blue dye (lymphazurine and patent blue). To be able to more accurately detect the amount of SLN and its anatomical location, a preoperative 3D single-photon emission tomography imaging is performed [115]. However, this technique consists of some limitations: (i) preoperative injection of radiopharmaceuticals is a painful procedure; (ii) blue dye can cause allergic reactions and stain at the injection site; and, (iii) when the lymphatic tissue is covered by skin or fat the visualization of blue dye is limited, resulting in a lower detection rate [116].

In recent years, SLN mapping using indocyanine green (ICG) with near-infrared fluorescence imaging has demonstrated feasibility and superiority when compared with blue dye, an easier application than TC-99, and better overall and bilateral detection rates when compared with TC-99 and blue dye [117,118,119,120,121], becoming a promising staging alternative in gynecological malignancies, and has been validated in patients with cervical and endometrial cancer [122,123]. Based on this evidence, studies have been published using ICG in patients with vulvar cancer [118,123,124,125,126,127,128,129,130,131,132,133]. Broach et al. described the largest cohort with SLN mapping using different techniques (TC-99 alone; blue dye alone; ICG alone; TC-99 and blue dye; TC-99 and ICG; TC-99, blue dye and ICG; and ICG and blue dye) with near-infrared imaging. One hundred and sixty patients with different histological subtypes of vulvar cancer were included. Of these, 114 patients had vulvar SCC, representing 195 groins at risk. Of the 195 at-risk groins, 25 (12.8%) were SLN positive and the two-year recurrence rate was 4%. Of the 169 negative groins, the two-year recurrence rate was 1.2%. Regardless of the histological subtype, the SLN detection rate by technique used was: for TC-99 and blue dye, 91.8% of the groins; for TC-99, blue dye and ICG, 96% of the groins; and, for TC-99 and ICG, 100% of the groins. With this, the authors concluded that SLN mapping in vulvar cancer is reliable and effective and that the use of ICG is associated with high SLN detection rates [134].

Siegenthaler et al. evaluated the SLN detection rate of ICG compared with TC-99 and blue dye. Sixty-four groins from 34 patients with vulvar cancer were analyzed. The 2-year groin recurrence rate was 2.9%. When analyzing SLN detection, the detection rate of ICG (87.5%) was similar to TC-99 (89.7%) and significantly higher than blue dye (77.8%). When combining TC-99 and ICG, the detection rate of SLN was 96.6%, being the best detection rate found in the study. In patients with lymph node metastasis or lymphatic vascular space invasion, ICG was significantly higher than TC-99 (*p* = 0.035 and 0.004, respectively). Finally, the authors concluded that SLN mapping using ICG with near-infrared fluorescence imaging in patients with vulvar cancer is feasible, accurate, and safe [116].

A European multicenter prospective observational study, GROningen International Study on Sentinel Nodes in Vulvar Cancer (GROINSS-V) evaluated the detection and clinical safety of SLN in vulvar cancer. Published results reported that SLN use is constantly increasing for the treatment of early-stage vulvar cancer [101]. This procedure aims to detect lymph node metastases in the SLN, avoiding performing a total lymphadenectomy in patients with negative SLN, therefore, reducing the morbidity associated with complete dissection of the inguinofemoral lymph nodes [13,101].

According to the GROINSS-V, the recommendations to indicate SLN biopsy are: (i) unifocal tumors confined to the vulva, (ii) vulvar tumors less than 4 cm in diameter, (iii) stromal invasion greater than 1 mm, and (iv) clinically and radiologically negative inguinofemoral lymph nodes [101].

Additionally, in the GROINSS-V study, 403 women with unifocal tumors confined to the vulva less than 4 cm in diameter, with stromal invasion greater than 1 mm, and clinically negative lymph nodes, were included. No additional lymphadenectomy was performed on the negative SLN. The false-negative rate of the SLN biopsy was 3%. The groin lymph node recurrence rates were 2.5% in patients with negative SLNs and 8% in patients with positive SLNs, with a 5-year follow-up. Disease-specific overall survival after 10 years was 91% for patients with negative SLNs and 65% for patients with positive SLNs. Still, surgical morbidity was significantly reduced [88]. Furthermore, the study showed that the size of metastases correlates with disease survival, as they reported 94.4% overall survival for metastases equal to or smaller than 2 mm and 69.5% for metastases larger than 2 mm [114].

The Gynecologic Oncology Group (GOG) has been studying several surgical and radiotherapeutic techniques in the treatment of early-stage vulvar cancer [13,103,113,135,136,137,138].

GOG protocols 74 [136] and 88 [137] were finished prematurely, as they were not successful in trying to replace inguinal lymphadenectomy with radiation therapy (GOG 88) or a less radical surgical procedure (GOG 74), as both protocols showed an increase in groin recurrence [136,137].

GOG protocol 173 was a prospective multi-institutional validation study that evaluated whether SLN biopsy could replace inguinofemoral lymphadenectomy and become the standard treatment for women with early-stage vulvar cancer. This protocol included 452 women in whom tumors ranged from 2 to 6 cm in diameter, and at least 1 mm in the depth of stromal invasion. All patients underwent lymphatic mapping, SLN biopsy, and inguinal femoral lymphadenectomy. Of the 452 women, 418 had at least one SLN identified, and of these, 132 women (31.6%) had an incidence of lymph node metastasis. The rate of lymph node metastasis was 26.4% in women with tumors < 4 cm and 40.9% in women with tumors 4 to 6 cm. Of 132 women with lymph node metastasis, 11 (8.3%) had false-negative results on SLN biopsy. When analyzing the data, the authors showed that in women with tumors < 4 cm the false-negative predictive value was 2% and for women with tumors 4 to 6 cm it was 7.4%. With this, they stated that inguinofemoral lymphadenectomy can be replaced by SLN biopsy in women with vulvar SCC < 4 cm [103].

The GROINSS-V study [101] and the GOG 173 protocol [103] established the SLN biopsy as the gold standard in the treatment of patients with early-stage vulvar SCC, unifocal tumors up to 4 cm in size, and unsuspected inguinal lymph nodes. SLN biopsy is correlated with lower perioperative morbidity and incidence of lymphedema compared with complete inguinofemoral lymphadenectomy [101,103]. Table 3 presents the main findings of these studies.

**Table 3 cancers-14-04184-t003:** Characteristics of the main vulvar cancer studies on sentinel lymph nodes and surgical margins.

Authors	Year	Aim	N	Outcomes
Van der Zee, et al. [101]	2008	To analyze the clinical utility and safety of SLN biopsy in early-stage vulvar cancer.	403	SLN biopsy in patients with early-stage vulvar cancer detects SLN metastases in SLN-negative patients, has a low groin recurrence rate, excellent survival, and decreases treatment-related morbidity.
Oonk, et al. [114]	2010	To evaluate the association of SLN metastasis size and disease survival risk in patients with early-stage vulvar cancer	260	Disease survival is related to the size of the SLN metastasis
Levenback, et al. [103]	2012	To evaluate whether SLN biopsy replaces inguinofemoral lymphadenectomy in patients with vulvar SCC.	452	SLN biopsy can replace inguinofemoral lymphadenectomy in patients with vulvar SCC.
Te Grootenhuis, et al. [88]	2016	To evaluate the long-term follow-up of patients undergoing SLN biopsy regarding recurrences and survival.	377	Patients with negative SLN have a good survival rate, but 36% of these patients and 46% of patients with positive SLN may have local recurrence. However, the surgical morbidity of these patients is significantly reduced.
Te Grootenhuis, et al. [72]	2019	To evaluate the incidence of local recurrence of vulvar SCC in relation to pathologic margins free of tumor and/or precursor lesion.	287	Local recurrences occur frequently in patients with primary vulvar carcinoma and are associated with dVIN at the pathologic margin, rather than any distance from the tumor-free margin.
Bedell, et al. [73]	2019	To analyze whether re-excision or adjuvant radiation in patients with early-stage vulvar cancer with a close or positive surgical margin improves recurrence-free survival.	150	Any additional treatment after primary surgical resection in patients with early-stage vulvar cancer did not show an improvement in local recurrence-free survival and overall survival rates, however, an improvement in the recurrence-free survival of these patients was observed.
Barlow, et al. [74]	2020	To analyze survival rates after conservative vulvar resection and determine clinicopathological predictors regarding vulvar recurrence, with a focus on surgical margin.	345	Treatment by re-excision or radiation therapy in positive or close margins (<5 mm) significantly decreases the risk of recurrence.

SCC: squamous cell carcinoma; dVIN: differentiated vulvar intraepithelial neoplasia.

For vulvar basal cell carcinoma, treatment consists of wide local excision and continuous follow-up because lymph node metastases are extremely rare [139,140]. In contrast, for lesions in the midline, bilateral lymphadenectomy is indicated [103]. In cases of vulvar melanoma, SLN biopsy is indicated for surgical resection of the primary tumor [141]. Evaluation of uni- or bilateral lymph nodes follows the criteria for vulvar SCC. In patients with verrucous carcinoma, the lesions are invasive, with reported cases of tumors measuring 15 cm with slight or no risk of lymph node metastasis. Nevertheless, due to the likely coexistence of verrucous carcinoma with SCC and their differences in treatment, an adequately large and deep biopsy should be performed to rule out concomitant disease. If SCC is excluded, lymph node dissection can be disregarded for verrucous carcinoma. For cases of vulvar sarcoma, lymph node dissection should be performed only for clinically positive cases. For Bartholin gland carcinoma, treatment recommendations are similar to vulvar SCC [43].

### 7.4. Adjuvant Treatment

Adjuvant treatment is a therapy provided after the primary treatment of a neoplasm and aims to reduce the risk of locoregional or extrapelvic recurrence in cases in which it was not possible to completely remove the tumor in surgery or the presence of lymph node metastasis. The mainstay of adjuvant therapy in vulvar cancer is radiotherapy with or without chemotherapy combination [142], and is indicated in cases of: (i) histologically confirmed lymph node metastasis, or, (ii) primary therapy for advanced-stage disease followed by radical resection of the residual tumor. However, there are other cases to consider, such as lymph vascular space invasion, deep invasion of the primary lesion, or large tumor size [70]. If SLN biopsy is positive, a complete bilateral inguinofemoral lymphadenectomy is recommended in cases of macrometastases (>2 mm), instead, in cases of micrometastases (>0.2 mm–≤2 mm), radiotherapy is sufficient [143]. If unilateral inguinofemoral lymphadenectomy was performed and the final pathology shows positive lymph nodes (macrometastases >2mm), contralateral inguinofemoral lymphadenectomy is recommended. In patients with inguinofemoral lymphadenectomy, the presence of two or more positive groin nodes or extracapsular spread, pelvic and groin radiotherapy is recommended [46,144,145]. In patients with excision margins < 8 mm, re-excision or radiotherapy is necessary [74].

Due to the rarity of vulvar cancer, prospective randomized trials of adjuvant therapy are particularly limited, and most approaches have been drawn from studies that have described heterogeneous treatments or studies of effective adjuvant therapies for cervical and anal cancer. In this context, the GOG 37 protocol included 144 women with positive inguinal lymph nodes after radical vulvectomy and bilateral inguinal lymphadenectomy to receive radiotherapy or pelvic lymph node resection. The two-year overall survival was 68% for patients who received adjuvant radiotherapy and 54% for patients who underwent pelvic lymph node resection [135]. In addition, long-term follow-up (mean 6 years) demonstrated a higher survival rate in patients who received radiotherapy (51% versus 41%) with significant benefit in patients with two or more clinically fixed ulcerated or positive inguinal lymph nodes [138]. This protocol demonstrated that the addition of adjuvant groin therapy and pelvic irradiation after radical vulvectomy and inguinal lymphadenectomy is effectively superior to pelvic lymph node resection and significantly reduces local recurrences and morbidity [135,138].

A large multicenter retrospective study, AGO-CaRE-1, was performed to better understand the role of adjuvant therapy in vulvar cancer node-positive. This study concluded that patients with positive inguinal lymph nodes who received adjuvant radiotherapy had an improved survival [145]. One of the retrospective subsets analyzed by the AGO-CaRE-1 study showed that radical groin lymph node dissection was similar to isolated sentinel lymph node dissection concerning recurrence and survival rates in lymph node-negative patients with a tumor measuring less than 4 cm [146]. In a current analysis of the AGO-CaRE-1 study, 306 patients were included according to criteria of nodal involvement, adjuvant radiotherapy treatment status, and radiation fields (groin/pelvis alone, vulva and groin/pelvis). It was observed that adjuvant radiotherapy in the vulva significantly reduces the risk of local recurrence in patients with positive lymph nodes. This observation was overexpressed in HPV-positive patients compared with the HPV-negative ones, but no relevant impact on the survival was observed. Accordingly, the HPV status is extremely important, and should be part of the pathological investigation since it is relevant to the therapeutic decision [147].

A multicenter cohort study published in 2020 analyzed the impact of adjuvant radiotherapy in women with single intracapsular lymph node metastasis. In this cohort, 176 women were enrolled, and it was observed that lymph vascular invasion was an independent risk factor related to shorter free survival. Adjuvant radiotherapy was an independent positive factor associated with recurrence-free survival. For this reason, the authors concluded that adjuvant radiotherapy in metastatic lymph node should be considered, especially in cases of lymph vascular invasion [148].

Van der Velden et al. analyzed the groin recurrence rate in 96 patients with a positive single intracapsular lymph node who did not receive adjuvant radiotherapy. After a median follow-up of 5 years, it was observed that 1% of patients were diagnosed with isolated groin recurrence and 2.1% of patients were diagnosed with a combined local and groin recurrence. The 5-year, disease-specific survival was 79% and overall survival was approximately 62%. The conclusion was that in such cases adjuvant radiotherapy can be safely omitted avoiding unnecessary toxicity and morbidity [149].

#### Role of Low-Volume Metastasis

The most important prognostic factor in vulvar cancer is lymph node metastasis [150]. In the last decade, the treatment of early-stage vulvar cancer has undergone major advances, in which SLN mapping should be highlighted [143,151,152,153]. SLN biopsy has become the standard of care in tumors less than 4cm instead of elective inguinofemoral lymphadenectomy and allows the search for low-volume metastases [154].

Reference publications evaluating the role of SLN showed that it is safe and accurate in early-stage vulvar cancer with tumors measuring less than 4 cm and unifocal disease [101,103] and reducing treatment morbidity [155].

The GROINSS-V II trial and the ongoing GOG 270 protocol (NCT01500512) compare groin radiotherapy with inguinal groin lymphadenectomy among patients with SLN metastases [143,152]. The GROINSS-V II study investigated whether complete inguinofemoral lymphadenectomy can be replaced by adjuvant radiotherapy in patients with early-stage vulvar cancer with low-volume metastases–SLN metastasis ≤ 2 mm. They also evaluated the safety, efficacy, and short- and long-term morbidity of lymphadenectomy versus adjuvant radiotherapy in patients with macrometastases (SLN metastasis > 2 mm). It was observed that inguinofemoral lymphadenectomy can be safely replaced by groin radiotherapy in patients with low-volume metastases, with minimal toxicity and morbidity [143].

## 8. Recurrence

In vulvar cancer, the risk of recurrence is associated with tumor size, lymph node involvement, and positive margins [156]. The overall recurrence rate is 37% at five years and most recurrences (40–80%) occur within two years of initial treatment [157]. For this reason, patients should be under medical surveillance after the end of treatment. Due to the rarity of this neoplasm, definitive data to inform the best surveillance strategy after treatment [158] is lacking, with this, the surveillance approach is extrapolated from experiences with cervical cancer [159]. According to the National Comprehensive Cancer Network (NCCN) Guidelines, and the Society of Gynecologic Oncology (SGO) recommendations, patients should be followed up with a physical examination every 3–6 months for the first 2 years, every 6–12 months for another 3–5 years, and then annually thereafter [90,160] (Table 4). Annual cervical and vaginal cytology examinations should be indicated to be able to detect any dysplasia in the female lower genital tract. However, because of the low probability of detecting recurrence, imaging tests (chest X-ray, CT scans, PET/CT scans, MRI) and laboratory tests (complete blood count, blood urea nitrogen, creatinine) should be indicated when there are symptoms of recurrence, clinical findings or suspicious test findings [90] (Table 4). A study published by Pouwer et al. showed that it is possible to detect asymptomatic isolated recurrences in the groin early, by following up on patients after treatment [161].

Recurrences are subdivided according to their location: isolated, groin-associated, or distant [162,163]. Local recurrences are located close to the site of the original vulvar lesion and are common in women with early-stage vulvar cancer with negative lymph nodes. Usually, this recurrence type is diagnosed more than two years after treatment with a median time of 33 months. In contrast, and not surprisingly, groin-associated or distant recurrences are frequent in women with advanced-stage vulvar cancer and/or with lymph node involvement and are diagnosed soon after treatment; the median time to groin-associated recurrence is 10.5 months and for distant recurrence is 8 months [164].

Decisions about the treatment of recurrent disease are often difficult and demand a tailored approach since it depends on the site of recurrence, previous treatment, and re-staging. Treatment includes surgery, radiotherapy, neoadjuvant chemotherapy, or palliative care [49]. In the case of local recurrence, the best conduct is surgical treatment (wide radical local excision and pelvic exenteration), since it has a good prognosis and survival rates of up to 60% in five years [87,165]. The treatment for groin-associated recurrence is surgery with or without radiotherapy, and has a 50% survival rate at seven years [96]. For distant recurrence, treatment is palliative since patients have a very poor prognosis [164].

Gonzalez Bosquet et al. performed a retrospective analysis of 330 women with primary SCC of the vulva who underwent lymphadenectomy. The recurrence rate in the first 2 years was 32.7% in patients with positive lymph nodes and 5.1% in patients with negative lymph nodes. Surprisingly, after 2 years, patients with positive lymph nodes and patients with negative lymph nodes had similar recurrence rates (~12%). However, 35% of the patients had recurrence after 5 years [157]. These results are similar to the long-term GROINSS-V I study. The GROINSS-V I study evaluated the long-term follow-up for recurrence and survival of 337 women with early-stage vulvar cancer who underwent the SLN procedure. The mean follow-up of these patients was eight years, reporting a high overall local recurrence rate of 27.2% at five years, and 39.5% at ten years after primary treatment. Ten-year disease-specific survival significantly decreased after local recurrence [88]. These results clearly prove the importance of long-term follow-up after treatment.

## 9. Conclusions

The incidence of vulvar cancer in young women is increasing every year due to HPV infection. For this reason, vaccination against HPV is very important since it is a prophylactic measure that can reduce the incidence of several neoplasms, including vulvar cancer. Vulvar SCC precursor lesions are classified into two distinct clinicopathological subtypes, uVIN and dVIN. The histopathological features of uVIN are more recognized than dVIN, and because of this, the diagnosis of dVIN may be delayed. Therefore, other genetic and epigenetic markers may be addressed to support the pathologist in accurately diagnosing these VIN subtypes. The suspicious lesion on the vulva should be biopsied and evaluated cautiously, since the lesions can resemble common dermatopathological diseases. The management of surgical treatment of vulvar cancer has progressed in recent decades to more tailored approaches. According to the NCCN and ESGO guidelines, a macroscopic surgical excision margin of at least 1 cm, which translates to a histopathological margin of 8 mm is recommended, as it decreases the risk of vulvar recurrences. The greatest progress in treatment has been the SLN biopsy, which is considered the gold standard in early-stage disease. The SLN biopsy can be performed in unifocal tumors confined to the vulva, less than 4 cm in diameter, stromal invasion greater than 1 mm, and clinically negative lymph nodes. Similarly, replacing lymphadenectomy with SLN mapping significantly has reduced surgical morbidity. However, more data are required to better delineate the optimal postoperative management of positive SLN. In this sense, we eagerly await the results of the GROINSS-V III trial that is evaluating the role of chemoradiation in patients with SLN macrometastases. Finally, local recurrence of the disease is very common and curable; however, groin-associated or distant recurrences have a worse prognosis. Therefore, it is indispensable to pursue a long-term follow-up after treatment.

## Figures and Tables

**Figure 1 cancers-14-04184-f001:**
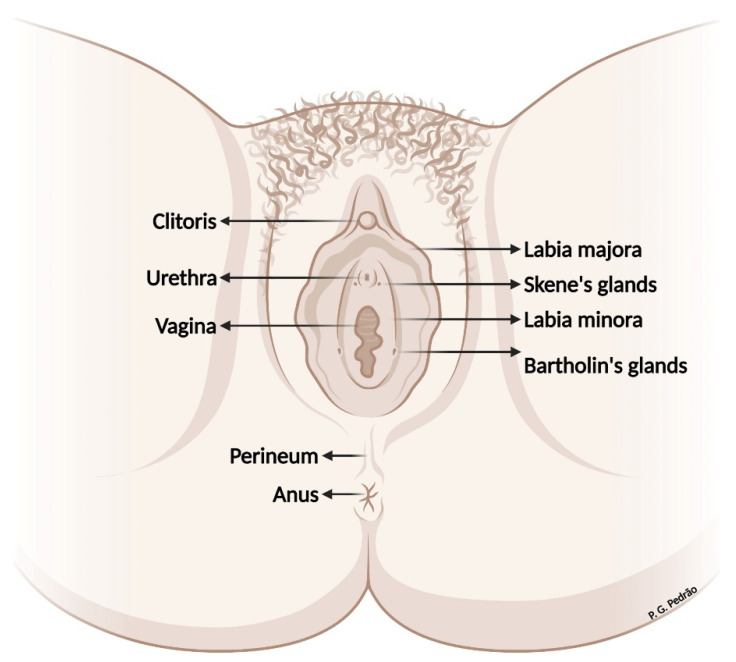
Vulvar anatomy.

**Figure 2 cancers-14-04184-f002:**
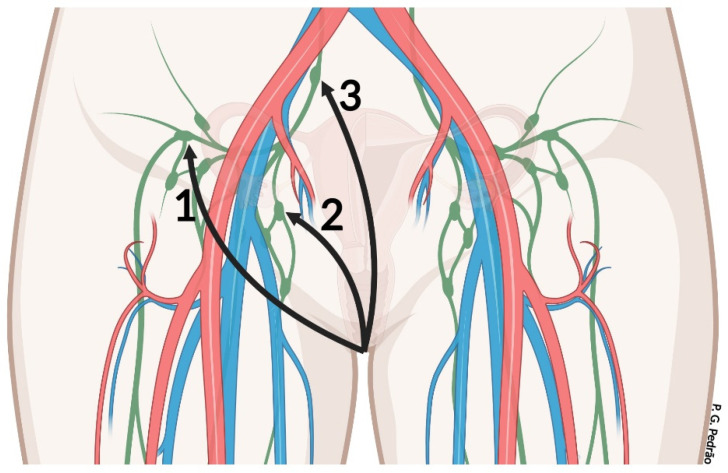
Lymphatic drainage on the vulva. The blue and red colors are the circulatory system, in which the blue color is the venous system and the red color is the arterial system. The green color is the lymphatic system. The numbers (1 to 3) indicate the inguinal lymph nodes of the vulvar lymphatic drainage. Number 1 is the superficial inguinal lymph nodes, number 2 is the deep inguinal lymph nodes, and number 3 is the iliac lymph nodes.

**Figure 3 cancers-14-04184-f003:**
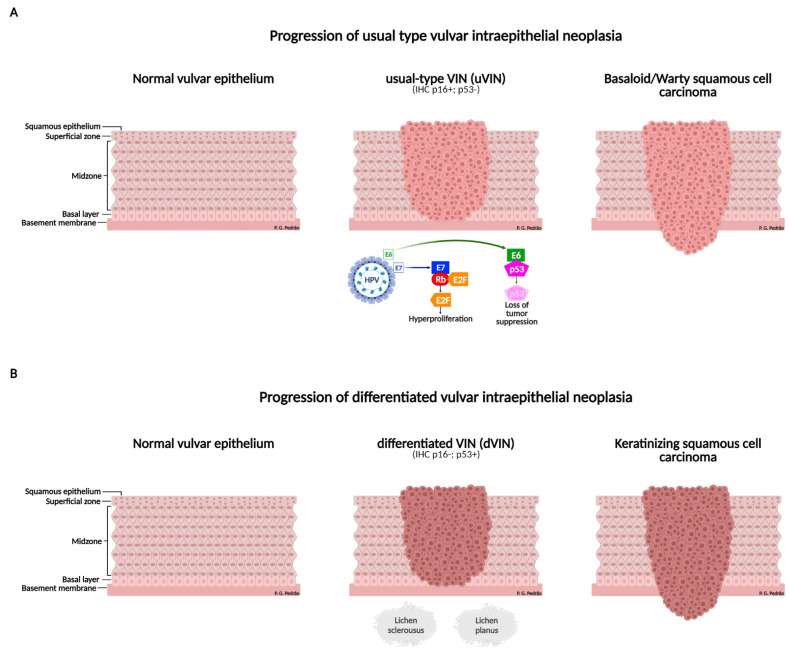
Development of squamous cell carcinoma of the vulva. (**A**) Progression of usual type vulvar intraepithelial neoplasia (uVIN). Usual VIN: associated with HPV infection, the E6 oncoprotein of the virus degrades the tumor suppressor protein, p53, and the E7 oncoprotein inactivates the retinoblastoma protein, Rb, which is also a tumor suppressor protein, releasing transmission factors, E2F, causing cellular hyperproliferation. In immunohistochemistry, the p16 biomarker is positive and p53 is negative. (**B**) Progression of differentiated vulvar intraepithelial neoplasia (dVIN). Differentiated VIN: it is not related to HPV infection, but arises through chronic dermatoses, mainly lichen sclerosus and lichen planus, which can progress to squamous cell carcinoma of the vulva. In immunohistochemistry, the p16 biomarker is negative and p53 is positive. uVIN: usual type vulvar intraepithelial neoplasia; dVIN: differentiated vulvar intraepithelial neoplasia; HPV: human papillomavirus; IHC: immunohistochemistry; VIN: vulvar intraepithelial neoplasia.

**Figure 4 cancers-14-04184-f004:**
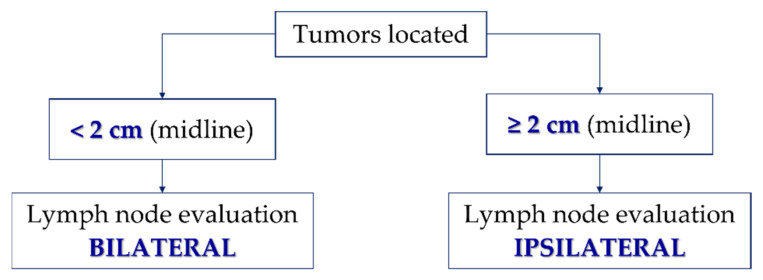
Flowchart of the evaluation of the lymph nodes about the location of the tumor about the midline.

**Figure 5 cancers-14-04184-f005:**
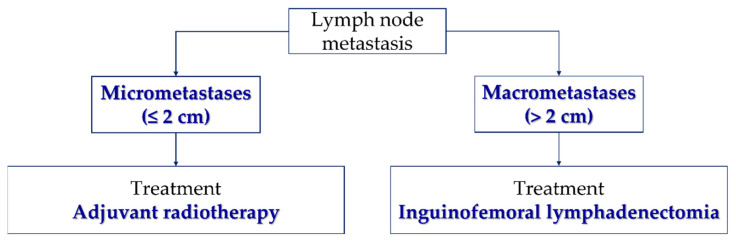
Flowchart of the treatment of metastatic lymph nodes.

**Table 2 cancers-14-04184-t002:** FIGO staging of vulvar carcinoma [49].

Stage	Description
I	Tumor confined to the vulva
IA	Tumor size ≤ 2 cm and stromal invasion ≤ 1 mm ^a^
IB	Tumor size > 2 cm or stromal invasion > 1 mm ^a^
II	Tumor of any size with extension to lower one-third of the urethra, lower one-third of the vagina, lower one-third of the anus with negative nodes
III	Tumor of any size with extension to the upper part of adjacent perineal structures, or with any number of non-fixed, non-ulcerated lymph nodes
IIIA	Tumor of any size with disease extension to the upper two-thirds of the urethra, upper two-thirds of the vagina, bladder mucosa, rectal mucosa, or regional lymph node metastases ≤ 5 mm
IIIB	Regional ^b^ lymph node metastases > 5 mm
IIIC	Regional ^b^ lymph node metastases with extracapsular spread
IV	Tumors of any size fixed to bone, or fixed, ulcerated lymph node metastases, or distant metastases
IVA	Disease fixed to pelvic bone or fixed or ulcerated regional ^b^ lymph node metastases
IVB	Distant metastases

^a^ Depth of invasion is measured from the basement membrane of the deepest, adjacent, dysplastic, tumor-free rete ridge (or nearest dysplastic rete peg) to the deepest point of invasion. ^b^ Regional refers to inguinal and femoral lymph nodes.

**Table 4 cancers-14-04184-t004:** Vulvar cancer surveillance recommendations [90,160].

Variable	Months		Years	
	0–12	12–24	3–5	>5
Physical examination	Every 3–6 months	Every 3–6 months	Every 6–12 months	Yearly
Papanicolaou test/cytologic evidence	Yearly ^a^			
Radiographic imaging ^b^	Insufficient data to support routine use			
Recurrence suspected	CT scans or PET/CT scans			

^a^ Insufficient evidence for detection of cancer recurrence but may have value in the detection of lower genital tract neoplasia and immunocompromised patients. ^b^ May include chest X-ray, positron emission tomography/computed tomography, magnetic resonance imaging, and ultrasound. CT: computed tomography; PET: positron emission tomography.

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
