# Peer review of "Management of Early-Stage Vulvar Cancer"

_cancers, 2022, doi:10.3390/cancers14174184_

Round 1

Reviewer 1 Report

Thank you for submitting the article to the journal.

The article is well-written, comprehensive, and current.

I have a few suggestions which would further enhance the article.

1. As a standard format, please include the article's objective in the introduction's last paragraph.

2. the Last name and first names on the first page are missing.

3. Please delete the vulvar melanoma paragraph.

4. Please include MRI and PET images of early-stage vulvar cancer.

5. There are no flowcharts. Based on the review, please make a master flowchart with the recommendations. 

6. Please add more tables. My suggestion, make a table of all-important studies and original articles, discussing essential conclusions.

Author Response

July, 14th, 2022

For Cancers

Referring: Reviewer 1

Comments and Suggestions for Authors

Thank you for submitting the article to the journal.

The article is well-written, comprehensive, and current.

I have a few suggestions which would further enhance the article.

  1. As a standard format, please include the article's objective in the introduction's last paragraph.

Answer: Thank you for the suggestion. In the last paragraph of the introduction, on lines 81-87, we have included the objective of the article.

  1. The Last name and first names on the first page are missing.

Answer: Thanks for the information, we have added it to the first page.

  1. Please delete the vulvar melanoma paragraph.

Answer: As a request, we have deleted the topic of vulvar melanoma.

  1. Please include MRI and PET images of early-stage vulvar cancer.

Answer: Thank you for the suggestion. We have added MRI and PET imaging, on lines 233-234, 674, and in table 4, line 680. However, imaging exams are more indicated in advanced tumors, in early-stage imaging exams are little applied.

  1. There are no flowcharts. Based on the review, please make a master flowchart with the recommendations.

Answer: Thanks for the suggestion. We have added two flowcharts on the recommendations for evaluation and treatment of metastatic lymph nodes (figure 4 line 427 and figure 5 line 430).

  1. Please add more tables. My suggestion, make a table of all-important studies and original articles, discussing essential conclusions.

Answer: Thanks for the suggestion. On line 557, a table (table 3) has been added with the results of the main vulvar cancer studies on sentinel lymph nodes and surgical margins.

Reviewer 2 Report

Thank you for this clear and comprehensible review on an important topic. These are my comments:

- The title is "management of early stage vulvar cancer": please define "early stage" (stage I and II? or tumor < 4cm?). Additionally several paragraphs also include information on advanced stage vulvar cancer. This should be clearly separated

- Line 55: I do not understand what does the 40.8% refer to?

- Figure 1: this illustration is rather redundant in my opinion. I would find an illustration of the lymphatic drainage more appropriate.

- Line 92: the lymphatic drainage from the clitoris is not necessarily made by the iliac region. To my knowledge it might be made by the iliac region.

- Line 130: How can uVIN account for 95% of all vulvar SCC cases when only 45% of the vulvar SCC are cause by HPV infection (Line 66)?

- Figure 2: I would recommend including the molecular p53 pathway in 2B. Furthermore the rearmost illustration of the carcinomas don't seem to not contain much additional information. To me they look the same. I would rather recommend to make an illustration of the precursor lesions and carcinomas highlighting their histological distictions

- In Chapter 4 I would add the more recent discussion on the third subgroup of p16 negativ and p53 negativ vulvar cancers (PMID: 33453182)

- Line 183: Please specify more in detail which imaging studies are recommended

- Figure 3: I would omit this figure.

- Line 204-206: Again, please specifiy which imaging you would suggest in which situation. 

- Line 365-366: To my best knowledge, GROINSS-V-II was not investigating the treatment options for the contralateral groin site

- Line 369-370: Not all of these patients need complete inguinofemoral lymphadenectomy, but all of them need a surgical lymph node staging (SLN or complete LNE)

- Line 403: Add PMID: 33968754 as a reference. It contains one of the largest cohort of ICG SLN mapping in squamous cell vulvar cancer patients to date.

- Line 422-432: For me the message of this paragraph is not clear

- Line 439-441: To my knowledge vulvar vasal cell carcinoma only need local excision as lymph node metastasis are extremely rare. PMID: 9351761, PMID: 31543771

- 7.4 adjuvant therapy: here is a lot of information according to advanced stage vulvar cancer. 

- Line 530-535: This statement was mentioned repeatedly

- Line 540-541: What is the rationale of follow up in vulvar cancer? Are there any studies demonstrating a survival benefit?

- 8. Recurrence: please add the latest findings on immunotherapy in advanced/recurrent vulvar cancer (Checkmate 358, Keynote 028, Keynote 158)

- 9. Vulvar melanoma: I would rather omit this subtype

Author Response

July, 14th, 2022

For Cancers

Referring: Reviewer 2

Comments and Suggestions for Authors

Thank you for this clear and comprehensible review on an important topic. These are my comments:

- The title is "management of early stage vulvar cancer": please define "early stage" (stage I and II? or tumor < 4cm?). Additionally several paragraphs also include information on advanced stage vulvar cancer. This should be clearly separated

Answer: Thank you for your comment. Yes, the early stage is stage I. According to all the authors, we don't think it is necessary to put "early-stage" in the title.

- Line 55: I do not understand what does the 40.8% refer to?

Answer: Thank you for your comment. The information has been adjusted in the text on line 60, in which, 40.8% refers to mortality in most high-income countries.

- Figure 1: this illustration is rather redundant in my opinion. I would find an illustration of the lymphatic drainage more appropriate.

Answer: Thank you for your comment. We would like to keep figure 1, and according to your suggestion, we made a new figure for lymphatic drainage (figure 2).

- Line 92: the lymphatic drainage from the clitoris is not necessarily made by the iliac region. To my knowledge it might be made by the iliac region.

Answer: Thank you for your comment. We don't understand your question, but on lines 98-105 we explain about lymphatic drainage, which begins in the superficial lymph nodes, followed by the deep lymph nodes, and finally the iliac lymph nodes.

- Line 130: How can uVIN account for 95% of all vulvar SCC cases when only 45% of the vulvar SCC are cause by HPV infection (Line 66)?

Answer: Thank you for your comment. We have corrected the information in the text, on line 72.

- Figure 2: I would recommend including the molecular p53 pathway in 2B. Furthermore the rearmost illustration of the carcinomas don't seem to not contain much additional information. To me they look the same. I would rather recommend to make an illustration of the precursor lesions and carcinomas highlighting their histological distictions.

Answer: Thank you for your comment. However, in our opinion, this would not add to the paper.

- In Chapter 4 I would add the more recent discussion on the third subgroup of p16 negativ and p53 negativ vulvar cancers (PMID: 33453182)

Answer: Thank you for your suggestion. On lines 155-158 we gave a brief introduction to the histology and immunohistochemistry of dVIN and uVIN, and then, on lines 158-165, we added the information from the suggested paper.

- Line 183: Please specify more in detail which imaging studies are recommended

Answer: Thank you for the suggestion. We have added MRI and PET imaging, on lines 233-234, 674, and in table 4, line 680. However, imaging exams are more indicated in advanced tumors, in early-stage imaging exams are little applied.

- Figure 3: I would omit this figure.

Answer: Thank you for your comment. We have removed the figure on mapping of lesions in the vulva, which was identified in figure 3.

- Line 204-206: Again, please specifiy which imaging you would suggest in which situation.

Answer: Thank you for your comment.  As previously answered, we report on the imaging exams on lines 233-234, 257, 674, and 680.

- Line 365-366: To my best knowledge, GROINSS-V-II was not investigating the treatment options for the contralateral groin site

Answer: Thank you for your comment. We make the purpose of the GROINSS-V II study clearer on lines 419-426.

- Line 369-370: Not all of these patients need complete inguinofemoral lymphadenectomy, but all of them need a surgical lymph node staging (SLN or complete LNE)

Answer: Thank you for your comment. But the information on lines 433-434 is from the FIGO 2021 paper.

- Line 403: Add PMID: 33968754 as a reference. It contains one of the largest cohort of ICG SLN mapping in squamous cell vulvar cancer patients to date.

Answer: Thank you for your suggestion. We thought it best to describe a bit about the conventional technique used in mapping so that we can then talk about ICG (lines 459-475). We added the study by Broach et al (2020) which was the largest cohort of SLN mapping with ICG (lines 475-486) and then added the study by Siegenthaler et al (2021) suggested by the reviewer (lines 487-496).

- Line 422-432: For me the message of this paragraph is not clear

Answer: Thank you for your suggestion. We have rewritten the paragraph making the information clearer (lines 538-550).

- Line 439-441: To my knowledge vulvar vasal cell carcinoma only need local excision as lymph node metastasis are extremely rare. PMID: 9351761, PMID: 31543771

Answer: Thank you for your comment. We have corrected the sentence according to the reviewer's information (lines 562-564).

- 7.4 adjuvant therapy: here is a lot of information according to advanced stage vulvar cancer.

Answer: Thank you for your comment. We only keep the information on the early stage of vulvar cancer.

- Line 530-535: This statement was mentioned repeatedly

Answer: Thank you for your suggestion. We have removed the information that was repeated.

- Line 540-541: What is the rationale of follow up in vulvar cancer? Are there any studies demonstrating a survival benefit?

Answer: Thank you for your comment. On lines 664-678 we describe post-treatment surveillance according to the NCCN and GBS. We have added a table (table 4) on line 679 about surveillance recommendations and on lines 703-715 we describe two studies that show the importance of long-term follow-up after treatment.

- 8. Recurrence: please add the latest findings on immunotherapy in advanced/recurrent vulvar cancer (Checkmate 358, Keynote 028, Keynote 158).

Answer: Thank you for your comment. The role of immunotherapy is currently growing, especially in the treatment of advanced endometrial cancer and advanced cervical cancer. We, the authors, believe that this topic should not be included in the chapter as we are dealing with early-stage vulvar cancer.

- 9. Vulvar melanoma: I would rather omit this subtype

Answer: As requested, we have removed the vulvar melanoma topic.

Reviewer 3 Report

This is a comprehensive review that should be of interest to clinicians.

If the following small changes are made it can be accepted.

-         1. Page 7, line 250: “radial” should be “radical”;

-          2. There are discrepancies in the way references are cited in the text (see lines 264, 278, 291, 318, 321, 324, 325, 326, 328, 507, 513). Please make the citations in the text more homogeneous;

-         3. page 13, line 553: are you sure you want to write "re-stagging" and not "re-staging"?

Author Response

July, 14th, 2022

For Cancers

Referring: Reviewer 3

Comments and Suggestions for Authors

This is a comprehensive review that should be of interest to clinicians.

If the following small changes are made it can be accepted.

  1. Page 7, line 250: “radial” should be “radical”;

Answer: Thank you for your comment. We have corrected the word (line 302).

  1. There are discrepancies in the way references are cited in the text (see lines 264, 278, 291, 318, 321, 324, 325, 326, 328, 507, 513). Please make the citations in the text more homogeneous;

Answer: Thank you for your comment, however, we found no discrepancy on lines 316, 329, 342, 368, 371, 374, 375, 376, 378, 629 and 635 as referenced by the reviewer.

  1. page 13, line 553: are you sure you want to write "re-stagging" and not "re-staging"?

Answer: Thank you for your comment. We have corrected the word (lines 694/695).

Reviewer 4 Report

Despite the extensive and up-to-date literature on various aspects of treatment of vulvar neoplasms in the internet resourcess, the undoubted value of the presented article is the comprehensive study of the topic. Considering the potential, broadly understood educational nature of the work, in the indroduction or in the part devoted to diagnostics in my opinion, a few sentences are missing about the differential diagnosis of tumor metastases from other locations to the vulva. 

Below are some of my other, little sugestions:

Line 149: Most vulvar cancer cases are diagnosed in the early stages...? 

the literature reffered to by the authors cvoncerns US, at this point I miss a short comment on the different stages of advancement found at the diagnostic in different countries.

Line 151: Vulvar cancer is asymptomatic in most women...?

I do not agree with this statement. At the any stage of invasive cancer  (specially these most common)  most pts more or less suffer from pruritus and /or pain and/or ulcer etc. The dominant type of symptom depend on advancement

Table 3: presented the new FIGO staging for vulvar carcinoma.  In my opinion it should be mentioned that this staging is applicable to most vulva malignancies except melanoma. It is advisable to supplement with staging for melanoma based on the depth ofthe invasion in the 5 part of article: Diagnosis

The text lines 218-228 are about diagnostics and therefore I suggest that it should be included in part 5.

Finnaly, just for the records , the treatment part is number 6, specific spectra of vulvar melanoma 7 and conclusions 8 rezpectively 

Author Response

July, 14th, 2022

For Cancers

Referring: Reviewer 4

Comments and Suggestions for Authors

Despite the extensive and up-to-date literature on various aspects of treatment of vulvar neoplasms in the internet resourcess, the undoubted value of the presented article is the comprehensive study of the topic. Considering the potential, broadly understood educational nature of the work, in the indroduction or in the part devoted to diagnostics in my opinion, a few sentences are missing about the differential diagnosis of tumor metastases from other locations to the vulva.

Below are some of my other, little sugestions:

Line 149: Most vulvar cancer cases are diagnosed in the early stages...?

the literature reffered to by the authors cvoncerns US, at this point I miss a short comment on the different stages of advancement found at the diagnostic in different countries.

Answer: Thank you for your comment. We have rewritten the paragraph (lines 179-183).

 Line 151: Vulvar cancer is asymptomatic in most women...?

I do not agree with this statement. At the any stage of invasive cancer  (specially these most common)  most pts more or less suffer from pruritus and /or pain and/or ulcer etc. The dominant type of symptom depend on advancement

Answer: Thank you for your comment. We have rewritten the paragraph (lines 189-193).

Table 3: presented the new FIGO staging for vulvar carcinoma.  In my opinion it should be mentioned that this staging is applicable to most vulva malignancies except melanoma. It is advisable to supplement with staging for melanoma based on the depth ofthe invasion in the 5 part of article: Diagnosis

Answer: Thank you for your suggestion. We added on line 245/246 that FIGO staging is applicable in most vulvar neoplasms except vulvar melanoma.

The text lines 218-228 are about diagnostics and therefore I suggest that it should be included in part 5.

Answer: Thank you for your suggestion. As suggested by the proofreader, the two paragraphs on lines 272-281 have been included in the diagnosis topic on lines 220-229.

Finnaly, just for the records , the treatment part is number 6, specific spectra of vulvar melanoma 7 and conclusions 8 rezpectively

Answer: Thank you for your comment. We have adjusted the numerical order of the topics.

Round 2

Reviewer 2 Report

Thank you for this interesting review. I only have a few comments:

Line 71: "approximately 95% of all vulvar SCC cases are thought to be caused by HPV infection": this is not correct. HPV prevalence is around 40% in vulvar cancer (PMID: 19115209, PMID: 23199955)

Line 103-104: "if the lesion is close to or on the clitoris, lymphatic drainage is made by the iliac region". This statement is not correct. According to the cited literature the lymphatic drainage from the clitoris is not necessarily made by the iliac lymph nodes. I would suggest to state: "if the lesion is close to or on the clitoris, lymphatic drainage might be made by the iliac region"

Line 149-153: this section is inconsequential: How can uVIN account for 95% of all vulvar SCC cases and dVIN for 35%? According to the cited literature (PMID: 28778635): "HPV-related SCC arises in younger women (63 years) and accounts for 20% of invasive disease versus non–HPV-related SCC (70 years), which accounts for 80% of invasive disease"

Figure 2: I would recommend including the molecular p53 pathway in 2B. Furthermore the rearmost illustration of the carcinomas don't seem to not contain much additional information. To me they look the same. I would rather recommend to make an illustration of the precursor lesions and carcinomas highlighting their histological distictions.

Figure 5 has a typing error: the first box should be "micrometastases" (< 2cm), not "macrometastases".

Author Response

July, 21sh, 2022

For Cancers

Referring: Reviewer 2

Comments and Suggestions for Authors

Thank you for this interesting review. I only have a few comments:

Line 71: "approximately 95% of all vulvar SCC cases are thought to be caused by HPV infection": this is not correct. HPV prevalence is around 40% in vulvar cancer (PMID: 19115209, PMID: 23199955)

Answer: Thank you for your comment.  Thank you for your suggestion. We have adjusted the sentence “In this context, approximately 45% of all vulvar SCC cases are thought to be caused by HPV infection.”

Line 103-104: "if the lesion is close to or on the clitoris, lymphatic drainage is made by the iliac region". This statement is not correct. According to the cited literature the lymphatic drainage from the clitoris is not necessarily made by the iliac lymph nodes. I would suggest to state: "if the lesion is close to or on the clitoris, lymphatic drainage might be made by the iliac region"

Answer: Thank you for your suggestion. We have adjusted the sentence on lines 104-105.

Line 149-153: this section is inconsequential: How can uVIN account for 95% of all vulvar SCC cases and dVIN for 35%? According to the cited literature (PMID: 28778635): "HPV-related SCC arises in younger women (63 years) and accounts for 20% of invasive disease versus non–HPV-related SCC (70 years), which accounts for 80% of invasive disease"

Answer: Thank you for your comment. We remove lines 150 to 154.

Figure 2: I would recommend including the molecular p53 pathway in 2B. Furthermore the rearmost illustration of the carcinomas don't seem to not contain much additional information. To me they look the same. I would rather recommend to make an illustration of the precursor lesions and carcinomas highlighting their histological distictions.

Answer: Thank you for your comment. We have improved figure 3 on the development of squamous cell carcinoma of the vulva.

Figure 5 has a typing error: the first box should be "micrometastases" (< 2cm), not "macrometastases".

Answer: Thank you for your comment. We have corrected the word in figure 5.

Thank you for your consideration.

Sincerely,

Ricardo do Reis, MD-PhD

Department of Gynecologic Oncology

Barretos Cancer Hospital, Pio XII Foundation, São Paulo, Brazil

Correspondence: [email protected]; Tel.: + 55-17-3321-6600 Ext 7126